# Increased Subcortical Sodium Levels in Patients with Progressive Supranuclear Palsy

**DOI:** 10.3390/biomedicines10071728

**Published:** 2022-07-18

**Authors:** Jannik Prasuhn, Martin Göttlich, Sinja S. Großer, Katharina Reuther, Britt Ebeling, Christina Bodemann, Henrike Hanssen, Armin M. Nagel, Norbert Brüggemann

**Affiliations:** 1Institute of Neurogenetics, University of Lübeck, 23562 Lübeck, Germany; jannik.prasuhn@neuro.uni-luebeck.de (J.P.); sinja.grosser@neuro.uni-luebeck.de (S.S.G.); katharina.reuther@neuro.uni-luebeck.de (K.R.); britt.ebeling@neuro.uni-luebeck.de (B.E.); christina.bodemann@neuro.uni-luebeck.de (C.B.); henrike.hanssen@neuro.uni-luebeck.de (H.H.); 2Department of Neurology, University Medical Center Schleswig Holstein, 23538 Lübeck, Germany; martin.goettlich@neuro.uni-luebeck.de; 3Center for Brain, Behavior and Metabolism, University of Lübeck, 23562 Lübeck, Germany; 4Institute of Radiology, University Hospital Erlangen, Friedrich-Alexander-Universität Erlangen-Nürnberg (FAU), 91054 Erlangen, Germany; armin.nagel@uk-erlangen.de; 5Division of Medical Physics in Radiology, German Cancer Research Center (DKFZ), 69120 Heidelberg, Germany

**Keywords:** ^23^Na-MRI, heteronuclear MRI, progressive supranuclear palsy, PSP, neurodegeneration

## Abstract

Progressive supranuclear palsy (PSP) is a debilitating neurodegenerative disease characterized by an aggressive disease course. Total and intracellular-weighted sodium imaging (^23^Na-MRI) is a promising method for investigating neurodegeneration in vivo. We enrolled 10 patients with PSP and 20 age- and gender-matched healthy control subjects; all study subjects underwent a neurological examination, whole-brain structural, and (total and intracellular-weighted) ^23^Na-MRI. Voxel-wise analyses revealed increased brainstem total sodium content in PSP that correlated with disease severity. The ROI-wise analysis highlighted additional sodium level changes in other regions implicated in the pathophysiology of PSP. ^23^Na-MRI yields substantial benefits for the diagnostic workup of patients with PSP and adds complementary information on the underlying neurodegenerative tissue changes in PSP.

## 1. Introduction

Progressive supranuclear palsy (PSP) is an atypical form of parkinsonism with a faster disease progression than PD [1]. However, in the early disease stages, the differential diagnosis between PSP and PD can be challenging. This is particularly relevant for certain PSP subtypes, e.g., PSP-P, where parkinsonism is partially levodopa responsive and PSP-defining features may be less prominent. An early diagnosis is thus desirable and reliable biomarkers are scarce [2].

Subsequently, a timely differential diagnosis often provides a significant challenge to the clinical practitioner, and instrumented diagnostics are desirable to support the diagnosis [3]. Structural magnetic resonance imaging (MRI) changes, e.g., an altered midbrain-to-pons ratio, occur only if a marked cell loss is already present and reliable biomarkers for the early disease phase are scarce [4]. Complementary biomarkers to structural atrophy patterns are desirable to address these challenges.

New imaging techniques are required to support an early diagnosis of PSP, which has led to the development of tau ligands for Positron Emission Tomography (tau-PET) [5]. However, tau-PET has the drawback of using ionizing radiation and is thus disadvantageous in the longitudinal assessment of patients. Furthermore, the specificity of tau ligands has been challenged due to non-target binding [6].

This study proposes using ^23^Na-MRI as a novel, non-invasive, and non-ionizing imaging modality to support the early detection of neurodegeneration and to improve our understanding of relevant disease mechanisms. In general, the fibrillary deposition of the tau protein impairs the correct assembly and overall stability of the Na^+^/K^+^-ATPase subunit complexes [7]. The Na^+^/K^+^-ATPase is a vital energy-dependent ion pump that removes intracellular sodium for extracellular potassium ions. The presence of fibrillary tau clusters results in the displacement of the neuron-specific α3-subunit of the Na^+^/K^+^-ATPase complex, thus reducing the neurons’ capability to control membrane depolarization and intracellular sodium levels [8]. Besides, the (α3-subunit-containing) Na^+^/K^+^-ATPase has been shown to play a pivotal role in tau fibril endocytosis and the prion-like cytosolic behavior of tau protein [7]. Here, we demonstrated a widespread total sodium (tNa) increase in neuroanatomical regions that have previously been implicated in the pathophysiology of PSP. The combination of total (tNa) and intracellular-weighted (inversion recovery, IR-Na) ^23^Na-MRI protocols allows indirect inferences on the underlying tissue injury as elevation of tNa levels can result from extracellular (e.g., by edema) or intracellular processes (e.g., by inflammatory pathways) [9]. Patients with PSP have been chosen as a suitable neurodegenerative model disease where established atrophy patterns have already been identified [4].

## 2. Methods

### 2.1. Demographics and Clinical Assessment

We enrolled 10 PSP patients with the following clinical subtypes: six patients presented with a classical Richardson’s syndrome (PSP-RS), three with primary parkinsonism (PSP-P), and one patient with PSP/corticobasal overlap syndrome (PSP-CBS). Additionally, we enrolled 20 age-and gender-matched healthy controls (HCs). We only assigned study participants to the PSP group if subjects met the clinical criteria for the highest diagnostic certainty available (as defined by the revised MDS Clinical Diagnostic Criteria) [3]. We surveyed demographic data, a standardized patient history, and the premedication state. Trained movement disorder specialists performed the standardized neurological examination following the PSPRS (PSP Rating Scale) protocol [10].

### 2.2. Ethics Statement

The ethics committee of the University of Lübeck approved this study. All participants gave written informed consent before enrollment in this study. Our study followed the regulations of the Declaration of Helsinki.

### 2.3. MR Data Acquisition and Analyses

MR data (T1, tNA, and IR-Na) were acquired and preprocessed as previously described [11]. We only included MRI datasets if no conflicting lesions or relevant comorbidities were present following neuroradiological review.

We used the SPM12 software package (University College London, Wellcome Trust Centre for Neuroimaging, London, UK, http://www.fil.ion.ucl.ac.uk/spm/, assessed on 1 July 2022) for the preprocessing and for the statistical analysis of sodium images. Preprocessing included the following steps: (1) Co-registration of sodium images to the T1 image employing a six-parameter rigid body transformation. (2) Spatial normalization of the T1 image to the Montreal Neurological Institute (MNI) standard template. (3) Spatial normalization of the sodium images using the estimated transformation from the previous step. (4) Spatial smoothing of the sodium images with an 8 mm FWHM Gaussian kernel. A voxel-wise mass univariate *t*-test was carried out to test for between group differences.

The voxel-based morphometry (VBM) analysis was performed using the CAT12 toolbox (Computational Anatomy Toolbox 12; http://dbm.neuro.uni-jena.de/cat/ accessed on 1 July 2022). The preprocessing pipeline incorporated the following processing steps: (1) tissue segmentation and (2) spatial registration to achieve voxel-wise correspondence across subjects, (3) resampling to a voxel-size of 1.5 × 1.5 × 1.5 mm^3^, and in addition (4) corrections for volume changes due to the registration (modulation).

We extracted mean signal intensities (tNa and IR-Na) and gray matter volumes in several regions of interest using the Neuromorphometrics (NM) brain atlas (Neuromorphometrics Inc., Somerville, MA, USA, http://www.neuromorphometrics.com accessed on 1 July 2022). Based on the PSP literature, we selected the following atlas regions as regions of interest (labeling according to the NM atlas) [12]: brainstem, caudate, putamen, pallidum, thalamus, midbrain, middle frontal gyrus, precentral gyrus, precentral gyrus medial segment, supplementary motor cortex, superior parietal lobule, superior temporal gyrus, cerebellum exterior, and cerebellar vermal lobules I-X. We combined left and right brain structures. This resulted in 14 brain regions.

### 2.4. Statistical Analyses

Differences in total tNA- and IR-Na-signals between PSP patients and HC were investigated by a random-effects analysis applying voxel-wise two-sample *t*-tests controlling for age, gender, and total intracranial volume. We corrected for multiple comparisons applying a topological FWE correction (FWE-level α = 0.05; cluster defining threshold *p* = 0.001). Our primary hypotheses initially restricted the stepwise analysis to the brainstem and the midbrain using a mask created from the respective Neuromorphometrics atlas regions. The number of voxels included in the analysis was n = 1303 (voxel size 4 × 4 × 4 mm^3^). The analysis was performed using the SPM12 toolbox. The mean sodium signal in the significant cluster was extracted and correlated to the PSPRS score (Pearson’s product-moment correlation).

We applied two-sample *t*-tests to test for significant between-group differences in tNa in the n = 14 regions of interest. Before this analysis, we regressed out the effects of age, gender, and total intracranial volume by multiple regression. A false discovery rate (FDR) procedure was applied to correct for multiple testing (Benjamini and Hochberg method). A Lilliefors test was applied to ensure that the data were normally distributed. Regions showing significant differences in tNA were also tested for IR-Na and gray matter volume differences.

### 2.5. Data Availability

The data that support the findings of this study are available on reasonable request from the corresponding author.

## 3. Results

### 3.1. Demographics and Clinical Assessment

The mean age did not differ between the PSP (67.3 ± 7.9 years) and the HCs group (68.7 ± 8.7 years) (*p* = 0.893). Both groups showed an equal sex distribution (HC: female: 9/19, male 10/19; PSP: female 5/10, male: 5/10). The PSP group had an intermediary disease duration (43 ± 16 months) and on average a moderate disease severity (PSPRS: 31.0 ± 16.9; scale from 0 to 100) (see Table 1).

### 3.2. Increased Midbrain Total Sodium in PSP Patients

PSP patients showed a significantly increased sodium concentration in the left midbrain compared to HCs (0.05 FWE-corrected at the cluster level; FWE-corrected cluster *p* = 0.006; cluster size k = 88; peak t = 5.34; peak MNI coordinates x = −6 mm, y = −19 mm, z = −16 mm). The cluster is localized in the left substantia nigra as shown in Figure 1A. The total sodium was significantly correlated to the PSPRS (Pearson’s rho = 0.778; *p* = 0.014; Figure 1B). We did not find evidence for abnormal IR-Na values nor abnormal gray matter volume in the midbrain cluster (Figure 1C,D). At an uncorrected level, however, we observed an increased intracellular-weighted sodium signal in the right substantia nigra comparing PSP patients with HCs (FWE-corrected cluster *p* = 0.071; cluster size k = 7; peak t = 3.87; peak MNI coordinates x = 9, y = −19, z = −13 mm).

The results from the ROI-based analysis are depicted in Figure 2. Three of 14 brain regions showed significantly increased total sodium (tNa) concentrations in PSP (two-sample *t*-test; 0.05 FDR-corrected p-threshold: 0.0069): the midbrain (*p* = 0.0004; t = 4.07), the brainstem (*p* = 0.0069; t = 2.92), and the pallidum (*p* = 0.0007; t = 3.83). We did not find evidence for aberrant intracellular-weighted sodium levels (IR-Na) nor gray matter volumes (Figure 2) in our PSP sample. According to a Lilliefors test, all data were normally distributed (all *p* > 0.05).

## 4. Discussion

PSP is a debilitating disorder, and pathophysiology-guided neuroimaging is needed to guide the timely diagnosis and enhance our understanding of relevant disease mechanisms. Patients with PSP may serve as a suitable neurodegenerative model disease to study complementary imaging modalities to map neurodegenerative processes in vivo.

Our results indicate that the increased tNa values were not driven by the rise of intracellular sodium, since the intracellular-weighted sodium signal was not elevated (IR-Na). These findings favor the interpretation of observed tNA changes as being caused by extracellular edema and sodium accumulation, a surrogate marker for overall impaired cell viability, and not as an indirect measure of tau pathology [9]. The observed changes in total sodium content in the absence of profound morphometric differences could point toward the potential of ^23^Na-MRI to map neurodegeneration before extensive atrophy already occurred. Therefore, total sodium changes may follow the same patterns implicated for morphometric differences in patients with neurodegenerative diseases. However, longitudinal studies would be needed to confirm this hypothesis by employing tNA and IR-Na-MRI, but could potentially foster the future differential diagnosis between parkinsonian disorders. These findings may point toward tNa-MRI as a more sensitive marker of neurodegeneration preceding the disease phase in which measurable atrophy occurs. However, the cross-sectional design of our study may limit the validity of this interpretation. In an extended case-control study, we have demonstrated the potential of IR-Na MRI in detecting pathophysiological changes in a patient suffering from a mutation in the alpha 3 subunit gene of the sodium/potassium pump (*ATP1A3*)*,* resulting in a disease named Rapid-Onset Dystonia Parkinsonism [11].

In a recent study applying high-field ^23^Na-MRI, similar changes were demonstrated in the midbrain of patients with Parkinson’s disease. Comparative studies are therefore needed to investigate whether observed changes are specific to distinct Parkinsonian disorders [13].

In general, ^23^Na-MRI can be performed on clinical standard 3 T MRI scanners but benefits from higher static magnetic field strengths, e.g., by means of higher spatial resolution. The necessary hardware setup (e.g., the need for a ^1^H/^23^Na head coil) can hinder the translation to clinical practice. The application of the IR-Na sequence advances the interpretability of the tNa sequence. However, the IR-Na sequence has a lower spatial resolution; a higher specific absorption rate (SAR), which limits the optimal choice of acquisition parameters; and a decreased signal-to-noise ratio (SNR) due to the selective compartmental suppression of the ^23^Na signal [7,9]. Even though our findings yield import implications for future studies, the widespread application of ^23^Na-MRI may be substantially hindered by the availability of MRI sequences, the costs of the necessary hardware, and the absence of standardized analytical approaches.

Besides some disadvantages, ^23^Na-MRI will provide deepened insights into specific disease mechanisms and can help in complimentary map neurodegeneration in vivo. However, the combination with other, in particular pathophysiology-orientated, neuroimaging modalities would be desirable to further enhance the biological interpretability of our findings [13]. Future studies in larger cohorts are necessary to ensure the reproducibility of our findings. It should also be noted that the clinically distinct subtypes of PSP may potentially result in distinct patterns of elevated tNA signals.

Interesting research aims for upcoming studies include, but are not limited to:the neuroimaging-based differentiation of PSP subtypes,the applicability of ^23^Na-MRI in the longitudinal assessment of patients with PSP (among other atypical parkinsonian syndromes), andthe evaluation of ^23^Na-MRI for the differential diagnosis of patients with idiopathic Parkinson’s disease from atypical parkinsonism.

Our findings support the future use of this method to map neurodegenerative changes in vivo and foster deepened insights into one’s individual disease mechanisms. In summary, we strongly encourage the use of this methodology in the study of neurodegenerative movement disorders.

## Figures and Tables

**Figure 1 biomedicines-10-01728-f001:**
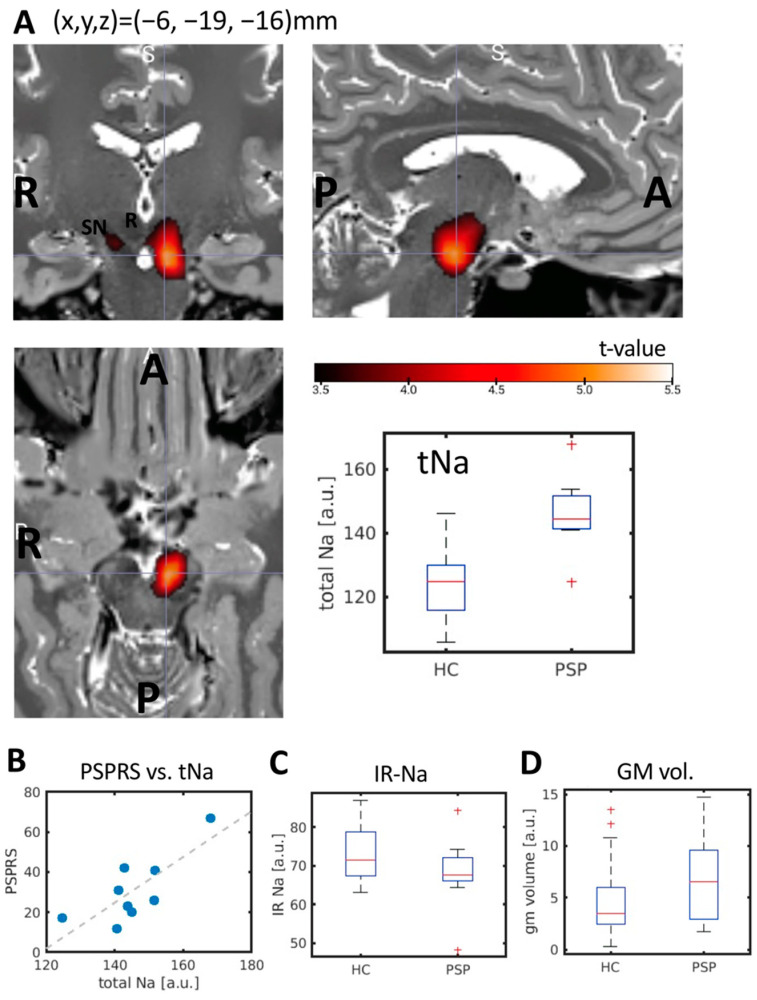
(**A**) Increased midbrain total sodium in PSP patients compared to healthy controls (0.05 FWE-corrected; significant cluster indicated by the crosshairs). (**B**) The total sodium was correlated to symptom severity as measured by the PSPRS (Pearson’s rho = 0.778; *p* = 0.014). We did not find evidence for significant between-group differences in (**C**) intracellular weighted sodium (IR-Na) nor (**D**) gray matter volume. a.u.: arbitrary units. GM: gray matter. HC: healthy controls. IR-Na: intracellular-weighted (inversion recovery) ^23^Na-MRI. PSP: progressive supranuclear palsy. PSPRS: PSP rating scale. R: red nucleus. SN: substantia nigra. tNa: total sodium weighted ^23^Na-MRI.

**Figure 2 biomedicines-10-01728-f002:**
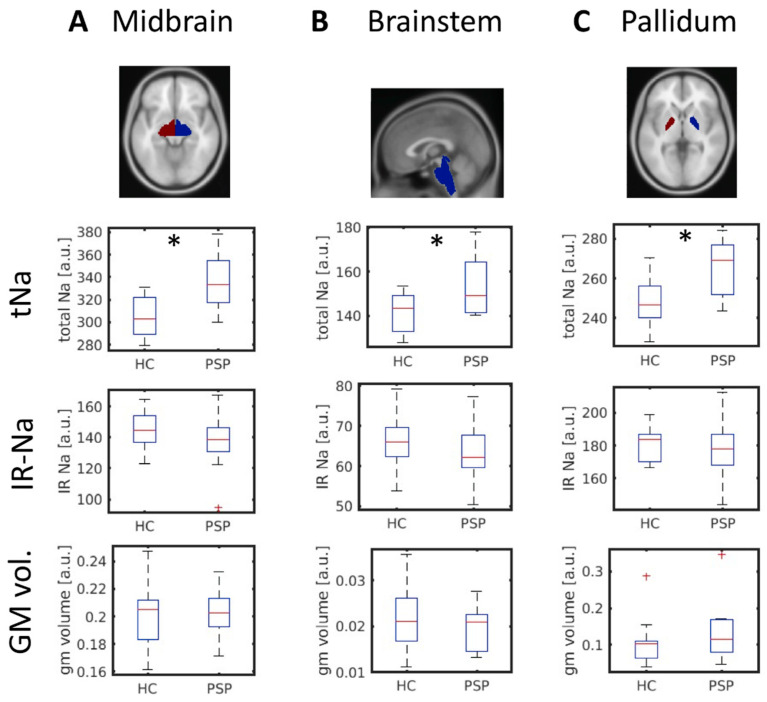
Brain regions showing an increased total sodium (tNa) concentration comparing PSP patients and HC (FDR-corrected). Shown are regions of interest according to the Neuromorphometrics Atlas ((**A**): Midbrain, (**B**): Brainstem, and (**C**): Pallidum; top row) and the tNa, IR-Na, and the gray matter volume in the respective regions. a.u.: arbitrary units. GM: gray matter. HC: healthy controls. IR-Na: intracellular-weighted (inversion recovery) ^23^Na-MRI. PSP: progressive supranuclear palsy. tNa: total sodium weighted ^23^Na-MRI. *: *p* ≤ 0.05.

**Table 1 biomedicines-10-01728-t001:** Demographic and clinical characteristics.

	PSP	HC	*p*-Value
number	10	19	
male/female	5/5	10/9	0.893 ^1^
age [years]	67.3 ± 7.9	68.7 ± 8.7	0.679 ^2^
PSPRS	31.0 ± 16.9 ^3^	n.a.	
MDS-UPDRS-III	36.1 ± 18.7 ^3^	n.a.	

Mean values and standard deviations are quoted. ^1^ according to a *X*^2^-test; ^2^ two-sample *t*-test applied; ^3^ available only for n = 9 PSP patients. HC: healthy controls. MDS-UPDRS-III: Movement Disorders Society’s Unified Parkinson’s Disease Rating Scale (subscore III). PSP: progressive supranuclear palsy. PSPRS: PSP rating scale.

## Data Availability

The data that support the findings of this study are available on reasonable request from the corresponding author. The data are not publicly available due to containing information that could compromise the privacy of research participants.

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
