# Peer review of "Increased Subcortical Sodium Levels in Patients with Progressive Supranuclear Palsy"

_biomedicines, 2022, doi:10.3390/biomedicines10071728_

Round 1

Reviewer 1 Report

Dear authors, 

This is an interesting study trying to improve the study of neurodegeneration in PSP patients. I have some suggestions and comments: 

1. Introduction. Can you describe more in depth why you decide to use Na-MRI? You described in the discussion, but for the relevance of the study, you should include here. 

2. Methods. T-test is a parametric tests, did the hypotheses of normality and variance homogeneity passed? Please, make sure that you checked normality and variance homogeneity before applying the t-test or any parametric. Otherwise, you need to apply non-parametric tests. 

3. Discussion. I missed correlation with other articles/other diagnostic techniques. Please, review the bibliography and describe how your approach could improve or complement current techniques. 

Thank you very much. 

Author Response

Dear authors, 

This is an interesting study trying to improve the study of neurodegeneration in PSP patients. I have some suggestions and comments: 
>> We thank the reviewer for the positive evaluation and the helpful suggestions/comments to further improve our manuscript.

  1. Introduction. Can you describe more in depth why you decide to use Na-MRI? You described in the discussion, but for the relevance of the study, you should include here. 
    >> We agree with the reviewer that the underlying rationale for applying 23Na-MRI to study neurodegenerative diseases should be included in the introduction of the manuscript. We have reordered the respective paragraphs accordingly.
  2. Methods. T-test is a parametric tests, did the hypotheses of normality and variance homogeneity passed? Please, make sure that you checked normality and variance homogeneity before applying the t-test or any parametric. Otherwise, you need to apply non-parametric tests. 
    >> According to the Lilliefors test the data were normally distributed for all regions of interest and both experimental groups (all p>0.05). Please note, that we visualized the data in Figures 1 and 2 and did not observe any indications that the data were not normally distributed. Furthermore, the voxel-level analysis was carried out with SPM12 and employed a two-sample t-test (mass univariate analysis). For consistency, we decided to perform a two-sample t-test also in the ROI-based analysis. We added the following information to the manuscript:

Methods section:
“A Lilliefors test was applied to ensure that the data were normally distributed.”

Results section:
“According to a Lilliefors test all data were normally distributed (all p>0.05).”

  1. Discussion. I missed correlation with other articles/other diagnostic techniques. Please, review the bibliography and describe how your approach could improve or complement current techniques. 
    >> In this study, we have only compared the applicability of 23Na-MRI to standard T1-based morphometry. To our knowledge, comparisons to other neuroimaging modalities are currently lacking. We have stressed this limitation in the discussion of our manuscript:

Discussion section:
“However, the combination with other, in particular pathophysiology-orientated, neuroimaging modalities would be desirable to further enhance the biological interpretability of our findings.”

Thank you very much. 

Reviewer 2 Report

In this communication, the authors assessed the use of total (tNa) and intracellular-weighted (IR-Na) sodium imaging (23Na-MRI) as a clinical tool to map neurodegenerative changes in vivo in 10 adult patients diagnosed with subtypes of progressive supranuclear palsy (PSP), a debilitating motor disorder. Compared with 20 age- and gender-matched healthy controls, patients suffering from PSP were found to have increased tNa levels in the midbrain, brainstem and pallidum, regions that have been implicated in the pathophysiology of PSP.

This clinical study appears to be an original and potentially significant advance in our understanding of PSP.  Increased brain tissue sodium concentration has been reported in specific brain regions of various neurodegenerative diseases, such as Alzheimer’s disease, amyotrophic lateral sclerosis and Freidreich ataxia, but has not been evaluated in PSP until now. 

Although the authors suggests that sodium imaging could support the early diagnosis of PSP, they do not explain how this technology would help differentiate PSP from other neurodegenerative diseases that share changes in brain tissue sodium concentration. A comparison of the regional location of these changes should be presented (possibly in a table format, including pertinent references) and discussed.

In addition, the introduction should be reworded because although differentiating PSP from different forms of Parkinson Disease (PD) is a diagnostic challenge (line 27-31), this study does not present any results comparing sodium imaging of PSP patients with PD patients.  Nevertheless, the authors may want to compare and discuss their findings with a recent article about sodium imaging in PD (Grimaldi S, et al. Increased Sodium Concentration in Substantia Nigra in Early Parkinson's Disease: A Preliminary Study With Ultra-High Field (7T) MRI. Front Neurol. 2021 Sep 9;12:715618. doi: 10.3389/fneur.2021.715618.)

Author Response

In this communication, the authors assessed the use of total (tNa) and intracellular-weighted (IR-Na) sodium imaging (23Na-MRI) as a clinical tool to map neurodegenerative changes in vivo in 10 adult patients diagnosed with subtypes of progressive supranuclear palsy (PSP), a debilitating motor disorder. Compared with 20 age- and gender-matched healthy controls, patients suffering from PSP were found to have increased tNa levels in the midbrain, brainstem and pallidum, regions that have been implicated in the pathophysiology of PSP.

This clinical study appears to be an original and potentially significant advance in our understanding of PSP.  Increased brain tissue sodium concentration has been reported in specific brain regions of various neurodegenerative diseases, such as Alzheimer’s disease, amyotrophic lateral sclerosis and Freidreich ataxia, but has not been evaluated in PSP until now. 
>> We thank the reviewer for the positive evaluation and the helpful suggestions/comments to further improve our manuscript.

Although the authors suggests that sodium imaging could support the early diagnosis of PSP, they do not explain how this technology would help differentiate PSP from other neurodegenerative diseases that share changes in brain tissue sodium concentration. A comparison of the regional location of these changes should be presented (possibly in a table format, including pertinent references) and discussed.
>> We agree with the reviewer that the rationale for this assumption is missing. We have added the following statement to the discussion of our manuscript:

Discussion section:
“The observed changes in total sodium content in the absence of profound morphometric differences could point towards the potential of 23Na-MRI to map neurodegeneration before extensive atrophy already occurred. Therefore, total sodium changes may follow the same patterns implicated for morphometric differences in patients with neurodegenerative diseases. However, longitudinal studies would be needed to confirm this hypothesis by employing total- and intracellular-weighted 23Na-MRI but could potentially foster the future differential diagnosis between parkinsonian disorders.”

In addition, the introduction should be reworded because although differentiating PSP from different forms of Parkinson Disease (PD) is a diagnostic challenge (line 27-31), this study does not present any results comparing sodium imaging of PSP patients with PD patients.  Nevertheless, the authors may want to compare and discuss their findings with a recent article about sodium imaging in PD (Grimaldi S, et al. Increased Sodium Concentration in Substantia Nigra in Early Parkinson's Disease: A Preliminary Study With Ultra-High Field (7T) MRI. Front Neurol. 2021 Sep 9;12:715618. doi: 10.3389/fneur.2021.715618.)
>> We agree with the reviewer and have reworded the introduction section accordingly. Please also refer to our answer to the comment above. In addition, we included the following paragraph to account for the recently published study on patients with idiopathic Parkinson’s disease:

Discussion section:
“In a recent study applying high-field 23Na-MRI, similar changes have been demonstrated in the midbrain of patients with Parkinson’s disease. Comparative studies are needed to investigate whether observed changes are specific to distinct parkinsonian disorders (Grimaldi et al., 2021). ”

Reviewer 3 Report

The authors presented a short communication on the feasibility of the imaging method.  It is interesting report howether I have few thoughts.

First question about the design of the experiment - do the authors think that such a number of patients in the study is sufficient to make concrete conclusions? I don't mean just the number (by the way, what is the incidence of PSP, subtypes of each? - the introduction seems insufficient) but more the distribution, especially the representative of the last category is one person.

In fact, the study provided only one piece of information on sodium elevation, I think that for the application of the method to diagnosis it would be worthwhile to try to find out more parameters.... on what basis do the authors think that it will be an appropriate differentiating feature in relation to Parkinson's or other neurodegenerative disorders? Since we are talking about the future, what are the costs of implementation and how likely, according to the authors, is it to be introduced into standard procedures?

Minor

line 23 a period in the middle of a sentence

Author Response

The authors presented a short communication on the feasibility of the imaging method.  It is interesting report howether I have few thoughts.
>> We thank the reviewer for the positive evaluation and the helpful suggestions/comments to further improve our manuscript.

First question about the design of the experiment - do the authors think that such a number of patients in the study is sufficient to make concrete conclusions? I don't mean just the number (by the way, what is the incidence of PSP, subtypes of each? - the introduction seems insufficient) but more the distribution, especially the representative of the last category is one person.
>> We agree with the reviewer on the relatively small sample size. However, we were able to demonstrate profound changes in total sodium levels in our cohort. To account for this limitation, we have added the following sentence/paragraph to our discussion:

Discussion section:
“Future studies in larger cohorts are necessary to ensure the reproducibility of our findings. It should also be noted that the clinically distinct subtypes of PSP may potentially result in distinct patterns of elevated tNA signals.”

In fact, the study provided only one piece of information on sodium elevation, I think that for the application of the method to diagnosis it would be worthwhile to try to find out more parameters.... on what basis do the authors think that it will be an appropriate differentiating feature in relation to Parkinson's or other neurodegenerative disorders?

>> We agree that the information from sodium imaging is limited although inferences can be made from the intra- and extracellular compartments. In this regard, we would like to refer to our previously published paper where we were able to identify compartment-specific sodium changes in a carrier of a mutation in the gene encoding the alpha 3 subunit of the sodium/potassium pump. We agree that future research should combine 23Na-MRI with other MR modalities (e.g. DTI including free water imaging, neuromelanin, quantitative susceptibility mapping) to collectively improve the diagnostic properties of a multimodal approach and to find the relationship between the different modalities.

Discussion:

“In an extended case-control study, we have demonstrated the potential of IR-Na MRI in detecting pathophysiological changes in a patient suffering from a mutation in the alpha 3 subunit gene of the sodium/potassium pump (ATP1A3, resulting in a disease named Rapid-Onset Dystonia Parkinsonism) (Prasuhn et al., 2022).”

“However, the combination with other, in particular pathophysiology-orientated, neuroimaging modalities would be desirable to further enhance the biological interpretability of our findings”

Since we are talking about the future, what are the costs of implementation and how likely, according to the authors, is it to be introduced into standard procedures?
>> We have added. The following sentence to the manuscript to account for this remark:

“Even though our findings yield import implications for future studies, the widespread application of 23Na-MRI may be substantially hindered by the availability of MRI sequences, the costs of the necessary hardware, and the absence of standardized analytical approaches.”

Minor

line 23 a period in the middle of a sentence
>> We thank the reviewer for pointing this out. Respective changes have been made in the manuscript.

Round 2

Reviewer 2 Report

Line 208-210: Please provide reference(s).

Line 222-225: Insert the reference number [13] for (Grimaldi et al., 2022) after the first sentence of that paragraph.

Line 233-240: The same sentence is duplicated.

Author Response

>> We thank the reviewer for the in-depth review of our revised manuscript. Please find our point-to-point response below.

Line 208-210: Please provide reference(s).

>> We have added the following reference for the interpretation of 23Na-MRI-derived signal alterations:

Huhn, Konstantin, et al. "Potential of sodium MRI as a biomarker for neurodegeneration and neuroinflammation in multiple sclerosis." Frontiers in neurology 10 (2019): 84.

Line 222-225: Insert the reference number [13] for (Grimaldi et al., 2022) after the first sentence of that paragraph.

>> We have added the required reference as an in-text citation to the manuscript and to the bibliography.

Line 233-240: The same sentence is duplicated.

>> We thank the reviewer for pointing this out and have removed the duplicated sentence.

Reviewer 3 Report

Authors adressed all concerns

Author Response

>> We thank the reviewer for handling our revised manuscript and are glad that we have resolved all raised concerns.